# Characterizing Powdered Activated Carbon Treatment of Surface Water Samples Using Polarity-Extended Non-Target Screening Analysis

**DOI:** 10.3390/molecules27165214

**Published:** 2022-08-16

**Authors:** Susanne Minkus, Stefan Bieber, Thomas Letzel

**Affiliations:** 1Urban Water Systems Engineering, Technical University of Munich, 85748 Garching, Germany; 2Analytisches Forschungsinstitut für Non-Target Screening (AFIN-TS) GmbH, 86167 Augsburg, Germany

**Keywords:** hydrophilic interaction liquid chromatography polar trace organic compounds, high-resolution mass spectrometry, sample comparison for process evaluation, principal compound analysis, volcano plot

## Abstract

Advanced wastewater treatment such as powdered activated carbon (PAC) reduces the load of organic micropollutants entering the aquatic environment. Since mobile and persistent compounds accumulate in water cycles, treatment strategies need to be evaluated for the removal of (very) polar compounds. Thereby, non-targeted analysis gives a global picture of the molecular fingerprint (including these very polar molecules) of water samples. Target and non-target screening were conducted using polarity-extended chromatography hyphenated with mass spectrometry. Samples treated with different types and concentrations of PAC were compared to untreated samples. Molecular features were extracted from the analytical data to determine fold changes, perform a principal component analysis and for significance testing. The results suggest that a part of the polar target analytes was adsorbed but also some byproducts might be formed or desorbed from the PAC.

## 1. Introduction

Chemicals of emerging concern (CECs), i.e., pharmaceuticals, chemicals of daily usage and others, enter the aquatic environment frequently through municipal or industrial wastewater and have already been detected in drinking water at ng L^−1^ levels [1,2]. In urbanized areas in Germany, the contributions of wastewater effluents to streams can in some cases be >50% in the summer months [3]. The de facto reuse of wastewater could increase the risk of introducing CECs into drinking water sources. Conventional water treatment processes have been reported to ineffectively remove certain CECs, such as hydrophilic compounds [1]. Measures that aim at further reducing the discharge of CECs include upgrading wastewater treatment plants (WWTPs) with advanced treatments such as ozonation or powdered activated carbon (PAC), a strategy currently pursued by Switzerland [4,5].

PAC is commonly used for drinking water purification due to its adsorptive qualities: in batch experiments, Hernández-Leal et al. treated ultrapure water spiked with CECs of personal care products (20–1600 µg L^−1^) with PAC at a dose of 1.25 g L^−1^ and found the removal efficiency to be >94% for all compounds after 5 min of contact time [6]. The log K_OW_ values of the investigated CECs ranged from −0.16 to 6.90 [7]. Kovalova et al. showed that 23 mg L^−1^ of PAC (hydraulic residence time of one day) removed 62% of the load of 56 CECs including pharmaceuticals, metabolites and industrial chemicals from hospital wastewater pre-treated by a membrane bioreactor. In their investigations, all compounds with a log *D* > 2 (pH 9) were eliminated completely or fell below the limit of quantification [8]. Adsorption capacities of polar CECs are expected to be lower than those of non-polar or mid-polar ones, in some cases up to one order of magnitude [9]. However, depending on the pH, some charged or zwitterionic polar compounds exhibit strong sorption, probably due to electrostatic interactions [8]. 

The occurrence and fate of CECs throughout treatment processes are typically assessed via targeted analyses. Focusing on a limited number of prioritized or probably already regulated target analytes could cause a partially distorted view of the molecular fingerprint of water samples as unknown or unexpected CECs are missed. At this point, it cannot be determined whether a decrease in concentration of a target analyte indicates its removal or merely its transformation. This leads to transformation products (TPs) being underestimated, even though they tend to be more polar (and thus mobile) and might even have enhanced toxicity potential compared to their parent compounds [10,11,12,13]. The non-target screening (NTS) approach, powered by high-resolution mass spectrometry (HRMS) often in combination with liquid chromatography (LC), allows a more comprehensive assessment of CECs as it refrains from preselecting substances [14,15]. When evaluating NTS data, features characterized by their retention time (RT), mass and signal intensity are extracted from full scan HRMS data and further processed following two general objectives:Identification with the methodological consequence of prioritizing features based on, i.e., the amount and quality of information available and their environmental relevance [16,17,18].Bulk characterization by a non-discriminatory feature extraction workflow followed by statistical analysis [19].

The latter is a suitable method to globally evaluate water treatment processes, often based on features’ signal intensities, that gives indications of the removal of CECs and formation of TPs [20,21,22].

In this study, the removal efficiency of PAC for CECs from water samples is investigated by NTS. Therefore, three different types of PAC at three different concentrations are assessed in batch experiments using the non-target screening approach. The polarity range of the analytical method is extended by coupling reversed-phase (RP) LC to hydrophilic interaction liquid chromatography (HILIC) in series and screening a mass window for small molecules using HRMS equipped with electrospray ionization (ESI). It should be noted that this study does not provide adsorption kinetics but rather presents a non-target screening method that enables a comprehensive comparison of untreated and treated samples, explicitly considering (very) polar compounds.

## 2. Results and Discussion

In the following, the results of the target evaluation are presented considering the internal standards and the polar standard compounds. Moreover, features extracted before and after PAC treatment are globally assessed with the objective of getting indications on removal and/or formation of CECs. Therefore, fold changes (fc) and multiple hypothesis tests were interpreted. Variabilities throughout the measurement sequence were evaluated based on the feature lists of all samples including pooled QC samples.

### 2.1. Targeted Evaluation

A targeted search for the internal standards (*n* = 10, added prior to sample analysis) and the polar standard compounds (*n* = 10, added prior to PAC treatment) was performed on the ESI(+) data. The RT and mass precision as well as the mass accuracy of the RPLC-HILIC-HRMS system was evaluated on the internal standards. In the positive ionization mode, the precision was better than 1.5 % for RT (*n* = 35 injections) and better than 1.5 ppm for mass. The mass accuracy was <2.1 ppm for all compounds. One measurement (absorbent H118 at 7 mg L^−1^, third replicate) appeared to be an outlier in the three parameters mass, RT and signal intensity, and was therefore discarded. In negative ionization mode, precision values for RT and mass were <0.8 % and 1.5 ppm, respectively. The masses of the internal standards were detected more accurately than 2.0 ppm (*n* = 36 injections). Based on these findings, the instrumental setup performed reproducibly and accurately. 

Furthermore, the consistency interval for the fold changes of the signal intensities was validated on the internal standards. It was defined as −1.00 ≤ log_2_(fc) ≤ 1.00, which corresponds to 0.50 ≤ fc ≤ 2.00 [22]. Since the internal standards were spiked into the samples after treatment and prior to analysis, the log_2_(fc) was expected to fall into the consistency interval and be close to 0. In all of the sample comparisons in the positive ionization mode, all internal standards fell into the consistency interval with at least ±9 standard deviations (see Table 1). The same was observed in the negative ionization mode with at least ±8 standard deviations (see Appendix A). The polar standard compounds were spiked into the samples prior to PAC treatment. In the positive ionization mode, their mean log_2_(fc) decreased with increasing PAC load and the variability of fold changes increased (Table 1). It can therefore be assumed that some of the standard compounds absorbed onto the activated carbon. Figure 1 represents the log_2_(fc)-RT plots of the PAC type H121 at different concentrations (2, 7 and 30 mg L^−1^). The internal standards (orange circles) all fall into the consistency interval (dashed lines). The (very) polar standard compounds (blue diamonds) eluted earlier than 15.6 min have log *D* values ≤ 0.30 and are thus expected to be primarily retained by the HILIC column. At 2 mg L^−1^ of PAC H121, they did not show any increase or decrease. At 7 mg L^−1^, a decrease in signal intensity (log_2_(fc) < −1) was observed for famotidine and 2,4-diamino-6-(hydroxymethyl)pteridine, complemented by 2-aminopyridine and 3-pyridinemethanol at 30 mg L^−1^. The respective plots for PAC H118 and H120 are given in Appendix A. Only miglitol and acamprosate were detected in all measurements conducted in the negative ionization mode and their fcs did not suggest any removal. Other polar standard compounds were partially filtered out during data processing.

### 2.2. Non-Targeted Evaluation 

When processing NTS data for process evaluation (Section 3), rather than filtering for features that could readily be matched with real compounds, it is important to curate the data to be statistically reliable. Thus, the data need to be cleaned from noise, artefacts and redundant peaks. 

Regions of increased data density were observed in the total ion chromatogram (Appendix A), i.e., approximately RT 25.6 min, where non-target peaks accumulate. In these regions the probability for the algorithm to erroneously extract artefacts might be increased. As a result, peaks were filtered by shape and number of data points before further processing. However, parameter ranges had to be set relatively wide to avoid false negative features. As a result, the bands of high feature density persist even after completing the full data processing regime (Figure 2). This matter is arguably better resolved by adapting the chromatographic separation rather than on a data processing level. 

Chromatographic peaks which were identified as isotopologues or adducts of features were removed in order to reduce redundancy in the feature lists. An alignment method was chosen based on the iterative RANSAC algorithm [23,24]. It allows to correct for some RT deviations which were observed for some HILIC-influenced analytes in the past [18]. In an attempt to eliminate misaligned features, the list was searched for duplicates which were then merged to consensus rows. 

There are features which are only detected in one or the other sample, either because they were only present in one, their concentrations fell below the limit of detection or the feature was not extracted from the data. Missing values pose an issue in comparative non-target screening analysis, as they hamper the calculation of fold changes. Missing value imputation to replace zeros should be chosen with care as different univariate or multivariate analysis require different methods [25]. Instead, a gap-filling step was implemented into the data processing workflow that conducts a targeted search to retrospectively pick missed peaks or baseline from the raw data. 

The results were nine final feature lists for each ionization mode (three PAC types, three concentrations). For each feature list, the mean log_2_(fc)-value and the respective standard deviation were calculated and are listed in Table 2 for the positive ionization mode and Appendix A for the negative mode. In contrast to what was suggested by the target compounds, no general decrease in signal intensity was observed as result of PAC treatment. On the contrary, the portion of non-target features with a log_2_(fc) > 1 increased at higher concentrations to a maximum of 13.4% at 30 mg L^−1^ of PAC H121 (for PAC H118 and H120 at the same concentration please refer to Appendix A). This result is visualized in Figure 2, showing that the portion of predominantly HILIC-influenced features with RTs < 17 min, exhibits an increase in signal intensity rather than a decrease. This could be attributed to contaminations desorbing from the PAC or, even though less likely, formation of transformation products. For post-treatments with granulated activated carbon and PAC dosed onto a sand filter, formation of transformation products has been observed before, possibly due to biological degradation [20,21]. Nevertheless, variability in the gap-filling results due to a noisy baseline could impair a clear distinction of consistent and increasing features. Our partner laboratory investigated the adsorption characteristics of (among others) the same PAC types under comparable experimental conditions, except they did not include a HILIC separation into their analytical platform [26]. For samples treated with 7 mg L^−1^ PAC compared to untreated samples, they found median fold changes of 0.54, 0.49 and 0.69 for H118, H120 and H121, respectively. Since these values are in or close to the defined lower limit of the consistency interval, no or little adsorption can be derived from these findings as well. 

### 2.3. Further Statistical Analysis

A PCA was computed using the feature lists prior to alignment in order to determine how individual samples group together based on the normalized peak heights. As an unsupervised, multivariate method, it further served the purpose to examine the pooled QC samples which were measured at the beginning and the end of sequence, as suggested by Sangster et al. [27]. Figure 3 displays the score plot for the data acquired in the positive ionization mode. The equivalent for the negative mode is given in Appendix A. The first dimension (principal component 1, PC 1) explains 22.4% of the variation and the second dimension (PC 2) 16.5 %. Technical replicates are overall grouped closely together in both ionization modes. Samples subjected to different treatments are mostly separated on the first principal component whereas the H118 isotherm is here also separated on the second dimension. The samples of the H120 isotherm (green) are grouped together suggesting little treatment effects. It was expected that the QC samples cluster tightly together and thus the variability among the test samples reflects the differences caused by treatment effects. However, the samples (green crosses) are positioned far apart on the PC 1 axis. In summation, these findings indicate that the run order might be possibly responsible for some variability between measurements. That a gradual change throughout the analytical sequence rendered the results less reproducible cannot be fully excluded at this point. On the other hand, the feature lists were investigated by PCA prior to alignment and the recursive gap filling. Consequently, some of the observed effects might be softened later on in the data processing regime.

If the features’ signal intensities were significantly different between the untreated and the treated sample, a Welch’s test was performed on signal intensities. Since the probability of wrongfully rejecting a true hypothesis increases for multiple comparisons (here approximately 3000 features), *p*-values need to be corrected [28,29]. In this study, the Benjamini–Hochberg method for controlling the false discovery rate was applied [30]. Finally, an exemplary volcano plot for PAC H121 at a concentration of 30 mg L^−1^ was constructed and presented in Figure 4. The features’ negative log_10_-transformed and adjusted *p*-values were plotted against the corresponding log_2_-transformed fold changes. Features with adjusted *p*-values < 0.05 (horizontal dashed line) and log_2_(fc)-values < −1 and log_2_(fc) > 1 (vertical dashed lines) were considered to be of significant decrease and significant increase, respectively. Out of a total of 439 features that fell outside the consistency interval, 340 were statistically significant, of which four were annotated with standard compounds. This additional level of security substantiates the assumption that either compounds are newly formed or desorbed from the H121 PAC material. However, for PAC H118 only 13 significant features with decreasing or increasing feature intensities were detected (Table 2), even though the PCA suggested otherwise. Features with signal intensities close to the limit of detection as well as background signals deconvoluted by the gap-filling algorithm, tend to introduce variability into the data. Since the standard error factors into the denominator of the t-statistic, the t-value decreases and the degrees of freedom increase which leads to higher *p*-values. For this reason, fold changes need to be considered as a criterion besides significance testing.

## 3. Material and Methods

### 3.1. Chemicals

Ultrapure water and acetonitrile were obtained at LC–MS grade from Supelco (Darmstadt, Germany) and Honeywell (Morristown, NJ, USA). Ammonium acetate was purchased from Sigma-Aldrich (Seelze, Germany). Information on (internal) standard compounds is given in Appendix A of the electronic Appendix A. The polar standard compounds for spike-in (log *D* at pH 7 of 0.30 to −4.10) were obtained from Neochema (Bodenheim, Germany), handled in four stock solutions at 10 µg mL^−1^ in methanol and stored at −18 °C. The compounds that served as internal standards were purchased from Sigma-Aldrich (Seelze, Germany) and Dr. Ehrenstorfer (Augsburg, Germany). They were prepared in individual stock solutions at 1000 µM, except for sotalol (586 µM), vidarabine (337 µM) and monuron (970 µM). They were dissolved in acetonitrile (etilefrine, sotalol, chlortoluron and metobromuron), acetonitrile/water (50/50, *v*/*v*; 6-amino-1,3-dimethyl-5-(formylamino)uracil, vidarabine and chloridazon) or methanol (chlorbromuron, metconazol and monuron) and stored at 4 °C.

### 3.2. Samples

For bench-scale batch sorption experiments, a surface water sample was taken from a German reservoir and filtered with a glass fiber filter (type GF 9; Schleicher & Schuell GmbH, Whatman, NH, USA) The dissolved organic carbon (DOC) concentration of the sample was 6.1 mg L^−1^, the pH 8.02 and the conductivity 526 µS cm^−1^ at 25 °C. The sample was spiked with 12 polar standard compounds at a final concentration of 50 µg L^−1^ per compound prior to treatment. The sample was separated into aliquots and treated with three different types of PAC (Table 3) at different concentrations. For each of the three PAC types, four 1 L-batches were prepared in beakers, adding no carbon (blank), 2 mg, 7 mg and 30 mg to the surface water sample. The batches were stirred at 250 rpm for 4 h at room temperature. Aliquots of all the batches were passed through a syringe filter (GF and 0.45 µm cellulose acetate, Minisart NML; Sartorius, Göttingen Germany) and transferred to baked-out vials (450 °C, 2 h). Treated samples were spiked with internal standards to a final concentration of 5 µM per compound prior to LC–MS analysis. Aliquots of all the samples were combined at equal volumes and also spiked with internal standards. Two triplicates of the pooled quality control (QC) samples were measured at the beginning and the end of the sequence.

### 3.3. LC–MS Analysis

The LC setup consisted of a HILIC and a RPLC system coupled in series via a T-piece with a mixing frit (Upchurch, IDEX Europe GmbH, Erlangen, Germany) [31].

Each LC system (1260 Infinity series; Agilent Technologies, Waldbronn, Germany) consisted of a binary pump, an online degasser and a mixing chamber. The RP separation was carried out on a Poroshell 120 EC-C18 column (50.0 × 3.0 mm, 2.7 µm; Agilent Technologies). The mobile phase consisted of 10 mM ammonium acetate in aqueous solution and acetonitrile at volumetric ratios of 90/10 and 10/90. For the HILIC subsystem, a ZIC-HILIC column was employed (150.0 × 2.1 mm, 5 µm, 200 Å; Merck Sequant, Umeå, Sweden) and the mobile phase consisted of acetonitrile and water. Information on the gradients can be found in previous publications [32,33]. The injection volume was 10 µL. 

The chromatographic system was connected to an Orbitrap Exploris 120 mass spectrometer (Thermo Fisher Scientific GmbH; Dreieich, Germany) equipped with an electrospray ionization (ESI) source. The source was operated at spray voltages of 3.5 and −2.5 kV in the positive and negative modes, respectively. Sheath gas, auxiliary gas and sweep gas were set to 50, 8 and 0 (arbitrary units). The capillary temperature and the vaporizer temperature were set to 320 and 400 °C, respectively. In order to obtain NTS data, a mass range of 70–1000 Da was scanned at a resolution of 60,000 (full width at half maximum at *m/z* 200). MS2 spectra were acquired in the data-dependent acquisition mode at a resolution of 30,000 by employing a collision energy ramp of 15–45 eV. The four most abundant precursor ions were selected to trigger after one scan cycle and afterwards excluded for 7 s. 

### 3.4. Data Analysis

#### 3.4.1. Extracting Target Compounds

The targeted analysis was performed in MZmine 2 [23]. The internal standards as well as the polar standard compounds were extracted at a mass tolerance of 3 ppm and a RT tolerance of 5 min. For further confirmation, detected MS2 spectra were compared to experimental MS2 spectra stored in the mass spectral databases MassBank [34,35] and MassBank of North America (MoNA) [36]. For the retention times (RT) and masses of the internal standards, the precision values were calculated, expressing the closeness of observed values to each other. Additionally, the mass accuracy was determined, expressing the closeness of detected masses to theoretical masses. 

#### 3.4.2. Extracting aligned non-target feature lists

The non-targeted data analysis was performed in MZmine 2 [23] as well. Figure 5 schematically shows the workflow for extracting features from NTS data, filtering and bringing the feature lists of the treated and untreated sample together for comparative analysis. The workflow is shortly outlined in the following: 

First, masses were detected on MS1 and MS2 level whereas the RPLC as well as the HILIC retention intervals (5 min–33 min) were considered. Then, chromatograms were built and peaks deconvoluted using the ADAP algorithm (“Automated Data Analysis Pipeline”) [37]. Chromatograms were smoothed by applying a Savitzky-Golay filter. Peaks were filtered out if the number of data points, tailing factor and/or asymmetry factor fell outside a predefined range. Thereafter, isotopic patterns of singly charged molecular ions were detected and isotopic peaks as well as sodium, potassium and ammonium adducts were removed. The feature lists were aligned across the three technical replicates using the RANSAC (“random sample consensus”) aligner [23,24] and subsequently duplicates were removed. Finally, the features that were detected in less than three replicates were eliminated.

The cleaned-up feature lists of the treated and the untreated sample were subsequently aligned for comparative analysis. For features which were only present or detected in one of the two samples, the respective gap was filled by a recursive targeted search in the raw data. Feature rows whose gaps could not be filled were removed from the list. Finally, the features’ signal intensities (peak heights) were normalized using the internal standards. All the standard compounds contributed to the normalization factor, but were weighted based on the *m/z* and RT distance to the feature [38]. The final feature list was exported to comma-separated values (CSV) format. The processing steps along with the parameter settings and explanatory comments are listed in Appendix A. Some parameters were optimized in a previous study on two synthetic water samples containing environmentally relevant and polar standard compounds at different concentration levels [18]. Other parameters were adjusted to fit the requirements of the present study, given for example by a different type of high-resolution mass spectrometer or research objective. Mass and RT tolerances were set based on the targeted analysis of the internal standards as well as the spiked HILIC standards. The parameter settings were validated and if necessary optimized on the three blank samples with the objective of obtaining full recall of the standard compounds and reducing the total feature number as well as the processing time. 

#### 3.4.3. Further Statistical Analysis

The fold changes (fc) of the target compounds and features were calculated as the ratio of the mean signal intensities across the technical replicates in the treated and the untreated sample. For each comparison (e.g., H118 (7/0) mg), the mean fc and the standard deviation of all the features were calculated. The threshold values for signal increase and decrease in a feature after activated carbon treatment were defined at log_2_(fc) >1 and log_2_(fc) <−1, respectively [22]. Further statistical evaluations were performed in R (version 4.0.2) [39] and RStudio (version 1.3.959) [40]. A principal component analysis (PCA) was carried out for the three blank samples, the nine treated samples and the six pooled QC samples for each ionization mode. The data comprised normalized peak heights and “0” was put for missing values. The data were scaled and centered before being submitted to the *prcomp* function. Whether the feature intensities are significantly different between the treated and the untreated sample was evaluated using Welch’s *t*-test [41] and the Benjamini–Hochberg adjustment [30] at a significance level of 0.05.

## 4. Conclusions

Non-target screening was carried out in order to evaluate the treatment of surface water samples with powdered activated carbon. 

A target screening in the ESI(+) mode of the polar standard compounds, spiked into the samples prior to PAC treatment, indicated the reduction of famotidine, 2,4-diamino-6-(hydroxymethyl)pteridine, 2-aminopyridine and 3-pyridinemethanol. No reduction was observed for 4-(2-hydroxyethyl)morpholine when treated with PAC H120 in contrast to PAC types H118 and H121. 2,2,6,6-tetramethyl-4-piperidone was only reduced by PAC H118. According to criteria defined within this study, PAC treatment did not affect 1,3-dimethyl-2-imidazolidinone, ectoine, miglitol and N,N′-ethylenebisacetamide. According to the presented results half of the highly polar standard compounds (log *D* < 0.30 at pH 7) were (partially) removed by at least one of the investigated PAC types. However, more and quantitative data are necessary to substantiate these findings. 

Non-target screening data were evaluated using fold changes, a principal component analysis and multiple hypothesis testing. Derived from the results, features exhibit repeatable peak heights across technical replicates; however, the run order appeared to introduce some variability throughout the sequence of measurements. The mean fold changes of non-target features fall into the consistency interval, overall suggesting no elimination or formation. However, the variability increases with PAC concentration indicating removal or formation of individual compounds. The fraction of features with increasing signal intensities (fold change > 2.00) was elevated at the highest tested PAC concentration of 30 mg L^−1^ with up to 13% and the highest number of significant features (based on differences in their signal intensities) observed for PAC type H121. Either compounds were newly formed or desorbed from the PAC material. It should be noted that the results could be affected by the feature extraction and data clean-up process, especially the gap-filling step. 

For future reference, the priority features (showing significant increase or decrease in feature intensities) could be identified and their (eco)toxicological relevance assessed. The applied chromatographic technology widens the view of molecules observed by mass spectrometric non-target screening. Thus, adsorption processes (and polarities of adsorbed molecules) can be monitored in a better way and classified more sustainably. 

## Figures and Tables

**Figure 1 molecules-27-05214-f001:**
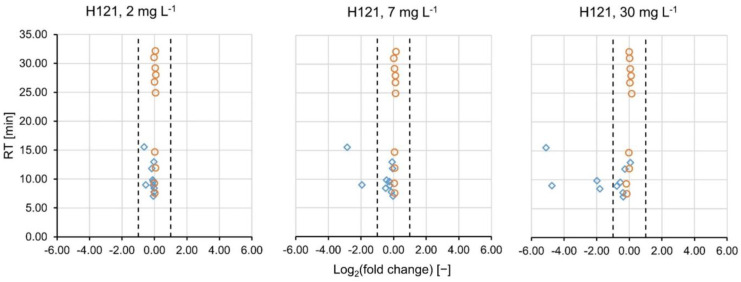
The base-2 logarithm of the fold changes of the HILIC standards (added prior to PAC treatment, blue diamonds) and the internal standards (added prior to analysis, orange circles) are plotted versus their RTs. The dashed lines mark the consistency interval where no compound removal is assumed.

**Figure 2 molecules-27-05214-f002:**
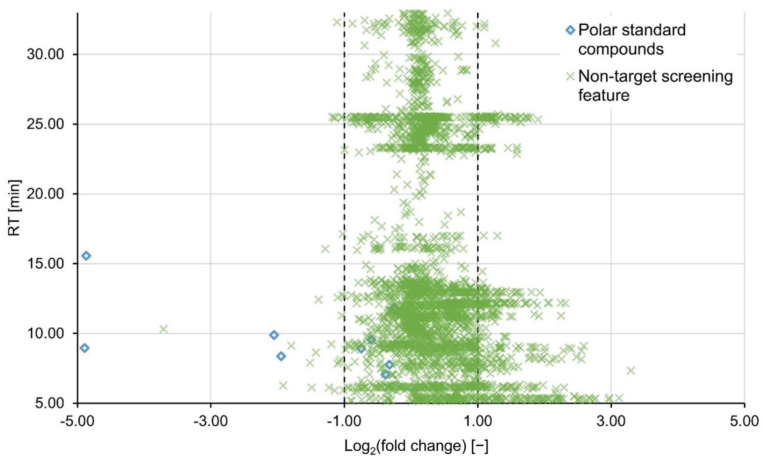
Non-target features (green crosses) and polar standard compounds (blue diamonds) are plotted by their base-2 logarithmic fold changes and RT. The dashed lines mark the consistency interval. Log_2_(fc) values <−1 and >1 are defined as a decrease and increase in signal intensity, respectively. Here, the sample treated with 30 mg L^−1^ of PAC H121 was compared to the untreated blank sample, both measured in the positive ionization mode.

**Figure 3 molecules-27-05214-f003:**
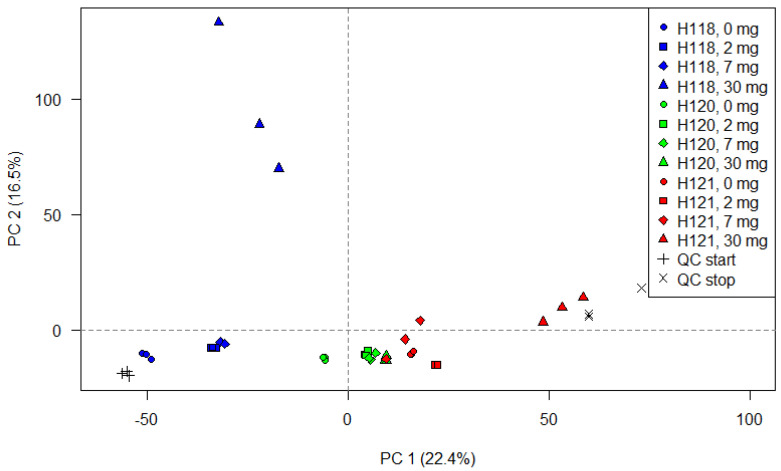
Scores plot of the PCA based on the normalized peak heights of the feature extracted from each individual measurement.

**Figure 4 molecules-27-05214-f004:**
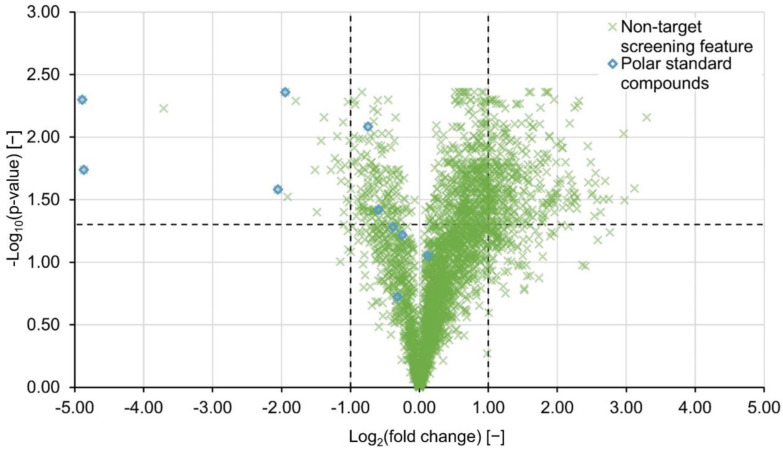
The volcano plot shows features extracted during the comparative analysis of the sample treated with PAC type H121 at 30 mg L^−1^ and the respective blank sample. The horizontal dashed line marks the cut-off *p*-value of −log10(0.05) and the vertical dashed lines the consistency interval. Features which were annotated with polar standard compounds are marked with blue diamonds.

**Figure 5 molecules-27-05214-f005:**
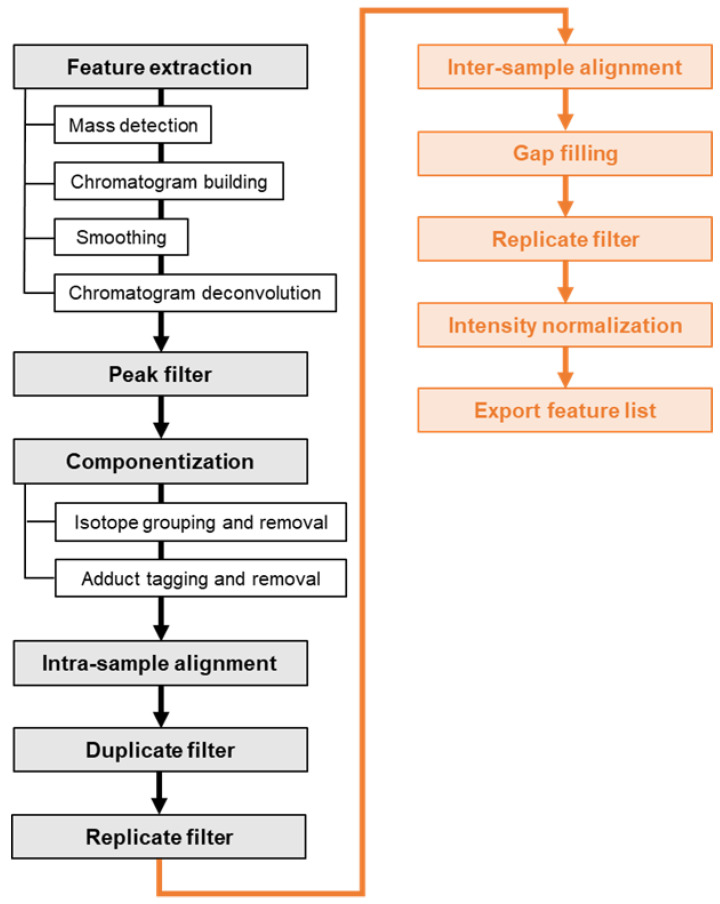
NTS data processing workflow for comparing a treated and an untreated surface water sample. The processing steps for extracting and filtering features from the technical replicates of a single sample are depicted in black. The orange boxes are processing steps that enable the comparative analysis of the treated and the untreated sample.

**Table 1 molecules-27-05214-t001:** Means and standard deviations of log_2_(fc) values for the internal standards and the polar standard compounds measured in the positive ionization mode. H118, H120 and H121 are the laboratory names of the different PAC types which were tested for surface water treatment at three different concentrations.

Concentration [mg L^−1^]	H118	H120	H121
	log_2_(fc)	log_2_(fc)	log_2_(fc)
**Internal standards**	*n* = 10	*n* = 10	*n* = 10
2	0.09 ± 0.05	−0.20 ± 0.07	0.03 ± 0.04
7	−0.04 ± 0.09	−0.25 ± 0.14	0.08 ± 0.04
30	−0.03 ± 0.10	−0.21 ± 0.16	−0.01 ± 0.10
**Polar standard compounds**	*n* = 10	*n* = 10	*n* = 10
2	−0.40 ± 0.49	−0.21 ± 0.49	−0.17 ± 0.22
7	−1.22 ± 1.46	−0.70 ± 1.30	−0.65 ± 0.95
30	−2.31 ± 2.21	−2.07 ± 2.45	−1.59 ± 1.88

**Table 2 molecules-27-05214-t002:** Means and standard deviations of log_2_(fc) values for the non-target features in the positive ionization mode. H118, H120 and H121 are the laboratory names of the different PAC types (Table 3) which were tested for surface water treatment at three different concentrations.

Concentration [mg L^−1^]	Number of Features	Mean log_2_(fc)	Increasing/Decreasing Features [%]	Significant Features
**H118**				
2	2981	−0.16 ± 0.38	0.6/2.7	38
7	3366	−0.26 ± 0.43	0.4/5.1	0
30	3000	0.07 ± 0.57	4.5/2.7	13
**H120**				
2	2941	0.11 ± 0.39	2.1/0.8	40
7	2856	0.11 ± 0.39	1.7/1.2	29
30	3058	0.17 ± 0.42	3.0/0.7	95
**H121**				
2	2842	−0.04 ± 0.37	1.1/1.8	28
7	2886	0.17 ± 0.41	3.5/0.7	2
30	3099	0.36 ± 0.62	13.4/0.8	336

**Table 3 molecules-27-05214-t003:** The types of powdered activated carbon (PAC) used for the batch sorption experiments. Values were partially published before [26].

Laboratory Name	H118	H120	H121
Manufacturer	Supplier A	Supplier A	Supplier B
Water content	8.1 %	1.5 %	2.0 %
Ash content	6.7 %	13.6 %	10.2 %
Contact pH	10.8	9.9	10.1
Iodine number	1088 mg g^−1^	1019 mg g^−1^	944 mg g^−1^
Particle size distribution (wet sieving)			
<150 µm	99.1	99.1	99.7
<50 µm	72.0	88.6	70.2

## Data Availability

Not applicable.

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
