# Peer review of "Characterizing Powdered Activated Carbon Treatment of Surface Water Samples Using Polarity-Extended Non-Target Screening Analysis"

_molecules, 2022, doi:10.3390/molecules27165214_

Round 1
Reviewer 1 Report
I regret that different process conditions (T, pH, time) at activated carbon application were not considered.
Author Response
Done.
pH was adjusted, because the pH value seems to be important
Reviewer 2 Report
In my opinion, the submitted manuscript is suitable for publication after a minor revision.
In my opinion, more information about the dangers and methods of removing pollutants should be added to the introduction.
I would also improve the quality of the figures. Unfortunately, not all of them are legible.
Overall, I rate the submitted manuscript positively.
Author Response
The quality of the figures 3 and 5 was improved as was also requested by reviewer 3.
Please see lines 36-48
Reviewer 3 Report
The manuscript entitled "Characterizing powdered activated carbon treatment of surface water samples using polarity-extended non-target screening analysis" is well studied and has sufficient innovation. The study is interesting and provides valuable information for the treatment of surface water samples using advanced wastewater treatment. The manuscript can be accepted as is, with minor revision. I think the following comments need to be addressed:
1. Keywords--Avoid duplicating words used in your title. Be specific and use multi-word “key phrases” where possible.
2. Line 26---authors cited old references. I suggest to add new references (https://doi.org/10.1016/j.envres.2021.112243; https://doi.org/10.1016/j.scitotenv.2021.148484).
3. Line 28, 40 … --There should be uniformity between numbers and units (for example, 50% or 50 %). Please check all and use same format throughout the manuscript.
4. Line 72---"according to proportions” …. Please cite Table 2 here.
5. In table 2 and 3, Kindly add heading to first column with units (as units are same “-L”).
6. Figure 3 and 5 could be improved. The black crosses in X and Y axes should be denoted by some color markings for clear vision. Please use full and don’t use abbreviation “fc” in the X-axis. “fc” should be replaced by “fold changes”.
7. Kindly use of abbreviation after the first appearance. For example, RANSAC. It needs to be reviewed throughout the manuscript.
8. Result and discussion: Explain better the connection between powdered activated carbon treatment and surface water. Kindly add more data referred to this in the discussion.
9. Conclusion section: Future developments is rather weak and short. What signification contribution this study to the society and social impact can also be discussed in this section. Kindly add future developments.
Author Response
|
Reviewer 3 |
|
|
1. Keywords--Avoid duplicating words used in your title. Be specific and use multi-word “key phrases” where possible. |
The keywords were adjusted. |
|
2. Line 26---authors cited old references. I suggest to add new references (https://doi.org/10.1016/j.envres.2021.112243; https://doi.org/10.1016/j.scitotenv.2021.148484). |
The paper suggestion sounds very interesting. Unfortunately, they are very specific and we would like to keep the introduction more general. |
|
3. Line 28, 40 … --There should be uniformity between numbers and units (for example, 50% or 50 %). Please check all and use same format throughout the manuscript. |
The spacing between numbers and units was adjusted. |
|
4. Line 72---"according to proportions” …. Please cite Table 2 here. |
We could not find the wording in our manuscript. |
|
5. In table 2 and 3, Kindly add heading to first column with units (as units are same “-L”). |
The tables were adjusted. |
|
6. Figure 3 and 5 could be improved. The black crosses in X and Y axes should be denoted by some color markings for clear vision. Please use full and don’t use abbreviation “fc” in the X-axis. “fc” should be replaced by “fold changes”. |
The figures were adjusted. |
|
7. Kindly use of abbreviation after the first appearance. For example, RANSAC. It needs to be reviewed throughout the manuscript. |
The explanation of the abbreviation was moved up within the manuscript. |
|
8. Result and discussion: Explain better the connection between powdered activated carbon treatment and surface water. Kindly add more data referred to this in the discussion. |
Done. The equivalents of Figure 3 for the other two PAC types were added to the supplementary material (Figures S3, S4) |
|
9. Conclusion section: Future developments is rather weak and short. What signification contribution this study to the society and social impact can also be discussed in this section. Kindly add future developments. |
done |